# Stigma as a barrier to addressing childhood trauma in conversation with trauma survivors: A study in the general population

**Georg Schomerus**[1]*, **Stephanie Schindler**[1], **Theresia Rechenberg**[1], **Tobias Gfesser**[1], **Hans J. Grabe**[2], **Mario Liebergesell**[1], **Christian Sander**[1], **Christine Ulke**[1], **Sven Speerforck**[1]

1 Department of Psychiatry and Psychotherapy, University of Leipzig Medical Center, Leipzig, Germany,
2 Department of Psychiatry and Psychotherapy, University of Greifswald, Greifswald, Germany

* Georg.schomerus@medizin.uni-leipzig.de

**Data Availability Statement:** All data files are available from the OSF database (accession number(s) https://osf.io/q4je7/.

## Abstract

Victims of childhood trauma report shame and anticipation of stigma, leading to non-disclosure and avoidance of help. Stigma is potentially aggravating the mental health consequences of childhood trauma. So far there is no comprehensive study examining stigma toward adult survivors of various forms of childhood trauma, and it is unclear whether stigma interferes with reaching out to affected individuals. In a vignette study based on a representative sample of the German general population ($N$ = 1320; 47.7% male) we randomly allocated participants to brief case vignettes pertaining to past childhood sexual/physical abuse or accidents, and adult physical abuse. Stigma was elicited by applying the Social Distance Scale, assessing respondents' attitudes/stereotypes toward the persons in the vignette and their reluctance to address the specific trauma in conversation. While one aim was to establish the prevalence of stigma toward persons with CT, we hypothesized that attitudes differ according to type of trauma. Of the respondents, 45% indicated they were unlikely to reach out to a victim of childhood sexual abuse, 38% to a victim of childhood physical abuse, 31% to someone reporting a childhood accident and 25% to someone reporting adult physical abuse. Contrary to our expectations, childhood sexual abuse did not consistently elicit more stigma than childhood physical abuse in Krukall-Wallis tests. Equally, childhood interpersonal trauma did not consistently elicit more stigma than childhood accidental trauma. Structural equation modeling revealed social distance as mediator of the relationship between negative stereotypes and reluctance to address childhood trauma in conversation. Our analyses further revealed an ambiguous role of negative stereotypes in addressing childhood trauma in conversation with trauma victims, which has yet to be examined. There is evidence for stigma associated with having survived childhood trauma, which is interfering with offering help.

**Funding:** The authors received no specific funding for this work.

**Competing interests:** The authors have declared that no competing interests exist.

## Introduction

Childhood trauma (CT) is a well-established causal risk factor for lifetime occurrence of mental disorders [1, 2]. Having suffered childhood maltreatment (i.e. physical, sexual or emotional abuse, or emotional or physical neglect) increases the risk of developing not only post-traumatic stress disorder, but several severe mental disorders such as depression [3, 4] borderline-personality disorder [5], substance use disorders, or schizophrenia [6]. CT has been described as a "hidden wound" carried on in later life, affecting the lives of victims for decades [1].

There is evidence that experiencing stigma is aggravating the consequences of CT [7]. Victims report shame, self-blame and anticipated stigma, leading to non-disclosure and avoidance of help [8]. Experienced and anticipated negative social reactions are associated with poorer outcomes like PTSD, depression and maladaptive coping [8]. In children, this has best been studied for child sexual abuse, where stigmatization (shame and self-blame) is a risk factor for adverse health effects [9]. In adults, negative consequences of stigma experiences have also been reported for women suffering from intimate partner violence [10, 11]. A study on long term effects of childhood sexual abuse on sexual difficulties found abuse-specific shame and self-blame, more than abuse severity, to be associated with later problems [12]. So far, emphasis has been put on survivors of sexual violence during childhood [8], or of intimate partner violence [10]. Stigma has been described as an "attribute that is deeply discrediting" [13, p 13]. By increasing social isolation, inducing shame, preventing help-seeking and overall spoiling the social identity of those having experienced CT [13], stigma may add to the adverse effects of CT on mental health. Indeed, previous studies confirm that shame dynamics are dominant peri-traumatic features of interpersonal trauma, and may play a dominant role in the development of PTSD [14].

Stigma is a social process. According to a widely used conceptualization, stigma consists of labeling, stereotyping, separation and status loss [15]. The prevalence of stigma on a population level has been shown to predict the degree of stigma experiences and self-stigma on an individual level [16]. Hence, although shame, self-stigma and anticipation of negative reactions occur within a stigmatized person [17], the root cause of stigma experiences is population attitudes [18]. The public stigma of CT could thus pose an area where some of the negative consequences of CT could be understood, addressed and prevented. Improving attitudes towards CT survivors could improve psychosocial outcomes of CT. However, to our knowledge, there are no comprehensive population-based studies examining public stigma towards survivors of CT, and the available studies do not differentiate between different types of trauma. Studies among students [19] and among judiciary clerks [20] examining the desire for social distance toward survivors of childhood sexual abuse found more than half of respondents reluctant to engage in various every-day situations. Other studies examined perceived blame attributed towards the society or the family [21] or differences in attitudes between victims and non-victims of childhood sexual abuse [19–22] either of these studies examined attitudes to different types of childhood trauma.

It would be important to understand to what extent the general population holds negative views of persons who have experienced CT, and if these differ between different forms of CT, and whether this indeed leads to reluctance to engage with someone with CT, should this person reach out for help.

In the current study, we examine three aspects of stigma and their interrelation: The respondents' desire for social distance toward adult victims of childhood trauma as an established measure of individual discrimination; the prevalence of negative stereotypes about someone who has experienced childhood trauma; and, as a potentially relevant specific

manifestation of victim stigma, the respondents' reluctance to reach out to adult victims of childhood trauma in conversation.

While our first aim is to establish the prevalence of stigma towards persons with different categories of CT, we further assume that attitudes differ according to the type of trauma. Our focus is on interpersonal trauma, specifically on childhood sexual and physical abuse, however, we add childhood accidental trauma and adult physical trauma as a control conditions. Broadly, interpersonal trauma can be distinguished from accidental trauma, and childhood trauma from adult trauma. Based on the literature on individual stigma experiences and from studies on trauma-related mental health problems [9, 23], we hypothesize that childhood sexual abuse elicits more stigma than childhood physical abuse (H1). Further, we hypothesize that childhood interpersonal trauma is associated with more stigma than childhood accidental trauma (H2). Since adults are perceived as more stable in order to overcome traumatic events, while children are regarded as more vulnerable and more profoundly affected by trauma, we assume that childhood physical trauma triggers more stigma than adult physical trauma (H3). Finally, to see to what extent the concept of stigma applies to public reactions to trauma survivors, we assumed that in accordance with established models of stigma [15], negative stereotypes about people with trauma experience increase the general desire for social distance from such persons, which ultimately manifests as a greater reluctance to talk to them about their trauma experience (H4).

## Materials and methods

### Sample

We conducted a population-based telephone survey (computer-assisted telephone interview, CATI), among persons aged 18 and older residing in Germany. Using a "dual-frame approach", the initial sample was randomly drawn from a combination of registered private telephone numbers and generated numbers, which allowed for the inclusion of ex-directory households, and from an additional share of mobile telephone numbers (30%). Target persons within households were selected randomly using the Kish-Selection-Grid. Persons who were reached via mobile telephone numbers automatically became target persons. Informed consent was considered to have been given when individuals agreed to complete the interview. The data collected in both modes are combined by a design weighting in which the probabilities of selection were mathematically corrected. Fieldwork was carried out by USUMA (Berlin), a company specialized in market and social research. The study was approved by the Ethics Committee of the Medical Faculty, University of Leipzig. In total, 1320 persons completed the interview, reflecting a response rate of 26.4%. The sample contained slightly more women and better educated persons than the general population (see Table 1).

### Vignette scenarios

At the beginning of the fully structured telephone interview, respondents were presented with a short vignette describing an encounter with a new neighbor, the gender of the person varying at random: *Imagine you have a new neighbor. When talking to you, they indicate that they have experienced [traumatic event] and are still dealing with the consequences.*

Respondents were randomly assigned to one of 4 versions of the scenario, differing solely by the [traumatic event] mentioned: sexual abuse as a child (n = 330), physical abuse as a child (n = 329), serious accident as child (n = 330), physical abuse as an adult (n = 331). By choosing these scenarios, we aimed to elicit differences in attitudes between sexual versus physical childhood abuse, interpersonal versus accidental childhood abuse, and childhood versus adult physical abuse. We chose "a new neighbor" to describe an encounter with a previously unknown

**Table 1. Socio-demographic characteristics of the study sample.**

| | | Total population of Germany [%] | Sample $N = 1320$ [%] |
|---|---|---|---|
| Gender | | | |
| | Male | 48.8 | 47.7 |
| | Female | 51.2 | 52.0 |
| | Diverse | n/a | 0.3 |
| Age | | | |
| | 18–29 | 17.0 | 10.8 |
| | 30–39 | 14.2 | 11.7 |
| | 40–49 | 19.9 | 13.9 |
| | 50–64 | 24.3 | 31.7 |
| | 65–74 | 13.5 | 15.8 |
| | 75+ | 11.2 | 16.1 |
| Educational attainment | | | |
| | 8–9 y | 35.1 | 15.3 |
| | 10 y | 23.6 | 28.6 |
| | >10 y | 41.1 | 56.1 |

Population data from the Federal Office of Statistics (Dec. 2019).

person that is relatable to all respondents and offers some choice as to how close a future relationship with the new neighbor will be. Generally, the use of vignettes allows the random assignment of study participants to different experimental conditions, which enhances the internal validity [24].

## Stereotypes

To assess to what extent the vignette [traumatic event] was linked to negative or positive stereotypes we elicited 10 potential stereotypes about a person having experienced the type of trauma mentioned in the vignette. These stereotypes were developed from discussions with people with lived experience and psychotherapists. Participating CT survivors and participating psychotherapists were asked about relevant positive and negative stereotypes in a semi-structured telephone interview. Statements started with "people who have experienced [traumatic event]. . .", followed by four positively framed statements (are able to have good friendships; are just as suitable for a responsible job like any other person; perform their parental duties just as well as other people; have survived a crisis and have grown through it), and six negative statements (are unpredictable; have a higher risk of becoming a criminal; are harmed for the rest of their lives; are guilty of what has happened to them to a certain degree; are not able to have a stable relationship; have already been vulnerable before the event, should they develop a mental illness). Answers had to be given on a 5-point Likert scale, ranging from "1" indicating strong agreement to "5" indicating strong disagreement. We inverted the scale so that the results may be interpreted intuitively with higher values representing stronger positive or negative stereotyping.

## Social distance scale

Following Phillips [25] who first employed a social distance scale in the context of a vignette experiment in 1963, and others (e.g. [26]), we used a 6-item social distance scale adopted from of an established social distance scale developed by Link and colleagues [26] and conceived by Bogardus [27], that has frequently been used in various vignette-based population studies in

Germany (e.g. [28]) as primary outcome. A good to excellent internal consistency reliability of social distance scales has been reported, ranging from 0.75 to greater than 0.90 [24, 29] and acceptable construct validity [24, 30]. The scale asks whether respondents are willing to tolerate the person described in the vignette in various hypothetical everyday situations with varying social distances: sublet a room, work together, take care of a young child, have married into family, introduce to friends, recommend for a job [26]. Items were rated on a 5-point Likert scale with 1 = "very likely" to 5 = "very unlikely". We calculated a mean score for all respondents who had answered at least 4 of 6 items ($N$ = 1305). Higher scores indicate a stronger desire for social distance. The desire for social distance is frequently used as an indicator of individual discrimination.

## Avoiding CT in conversation with CT victims

To elicit participants' avoidance to engage individuals with specific CT in conversation we also asked after having presented the vignette: *How likely would it be that you actively raise this topic again with your neighbor*? Answers were given on a 5-point Likert-scale ranging from 1 "very likely" to 5 "very unlikely". A total of 1313 responses were collected successfully. We inverted the scale to facilitate its interpretation, with higher scores indicating greater willingness to interact with a trauma survivor.

## Statistical analysis

Statistical analyses were performed with R version 3.6.3 and its package "stats" if not stated otherwise. In a first step, we describe the sample, frequencies of respondents' previous experience with CT, frequencies of knowing someone with CT, and avoiding CT in conversation, and the central tendency of the Social Distance Scale.

We then examine hypotheses H1 to H3, asking whether there are significant differences between the vignettes with regard to the level of stereotyping. We used the nonparametric H-test by Kruskal-Wallis as a global test with the significance level set to $p < 0.05$. In case of statistical significance this was followed by a pairwise comparison of the vignettes using Conover's test of mean rank sums with Holm's correction for multiple testing from the R-package "PMCMR". With the same approach, we examined whether desire for social distance differs between vignettes, and whether the reluctance to address different types of trauma in conversation differs according to the type of trauma mentioned. To reduce the information load of our report of these analyses, we collapsed the five categories of the stereotype items and the willingness to talk to a victim item into three categories representing "agree", "undecided", and "disagree".

Hypothesis H4, predicting that stigmatization of trauma victims results from negative stereotypes that motivate a general desire for social distance from such victims, was tested by a mediation analysis using structural equation modeling (SEM) with the R package lavaan [31]. The mediation model incorporated negative and positive stereotypes as latent exogenous variables (predictors), the desire for social distance as latent endogenous variable (mediator), and willingness to talk to a victim as manifest endogenous variable (criterion; please see S1 Fig for the model specification).

The manifest exogenous variables for the latent stereotype variables were assigned in accordance with the results of an ad hoc factor analysis. Specifically, we calculated a principal component analysis with all inverted 5-point stereotype items with acceptable Kaiser-Mayer-Olkin (KMO) values ($> 0.7$) using the R-package "psych". The item asking about being permanently harmed was excluded (KMO = 0.55). Two factors with Eigenvalues $> 1$ emerged, representing negative (Eigenvalue: 2.47) and positive (Eigenvalue: 1.46) stereotypes and cumulatively

explaining 44% of the variance. S1 Table lists the factor loadings for the two extracted factors calculated after varimax rotation using the R package "GPArotation". The stereotype indicators and all other indicator variables entered the SEM analysis as 5-point ordinal values and a weighted least squares mean and variance adjusted estimator was chosen.

In terms of global model fit we complemented the Chi$^2$ statistic, which is highly sensitive to sample size and model complexity [32], with the root mean square error of approximation (RMSEA; [33]), comparative fit index (CFI; [34]), and standardized root mean square residual (SRMR; [35]), as recommended by Kline (2016) [36]. There exists no clear consensus about the interpretation of descriptive fit indices, however. A normed Chi$^2$ / degrees of freedom of two and three has been considered a good and acceptable fit, respectively [32], whereas others discourage the use of this ratio [36]. A RMSEA and SRMR < .05 is currently accepted as good fit [32, 37, 38]. A CFI > .95, in some opinions > .97, is usually considered acceptable [32, 37, 38] although recent studies question the validity of such universal thresholds altogether (c.f. [36]).

We first tested hypothesis 4, represented by the mediation term–the product of coefficients, in the full sample (single-group SEM—SGSEM) using the conservative Sobel test [39]. The analysis was then repeated with vignette as a grouping variable to explore trauma-specific relationships (multi-group SEM—MGSEM). Missing values were deleted pairwise. Parameter estimates that were statistically significant also under listwise deletion of missing values were considered interpretable.

Finally, we tested metric and scalar measurement invariance for the measurement part of the structural equation model to determine whether (descriptive) differences between the path coefficients of certain groups may be related to psychometric non-equivalence of the three latent variables central to the model. According to Putnick [40] contemporary conventions consider a $\Delta$CFI $\leq$ -.01, $\Delta$RMSEA $\geq$ .01, and $\Delta$SRMR $\geq$ .015 as indicative of scalar measurement non-invariance.

## Results

### Evaluating stigma measures between scenarios (H1-3): Stereotypes

Table 2 shows the agreement with positive and negative stereotypes across vignettes. While most respondents agreed with positive, and disagreed with negative stereotypes, some patterns emerge: The most frequently endorsed assumption was that people are permanently harmed by their traumatic experiences. Of the respondents, 61.5% agreed with this statement for child sexual abuse, while only 14.8% disagreed. Agreement followed a clear gradient consistent with our hypotheses H1-H3: Childhood sexual abuse was seen most damaging (all pairwise comparisons $p < 0.005$), followed by childhood physical abuse (50.9%), adult physical abuse (42.9%) and childhood accident (30.9%).

Other stereotypes also showed differences between scenarios, but with varying patterns. People who experienced sexual abuse in childhood are regarded more frequently to be unable to have a stable relationship (endorsed by 24.1% of respondents), the difference being significant ($p = 0.021$), compared to the scenario mentioning a childhood accident (endorsed by 19.9%), and less frequently are they expected to have grown from their adverse childhood experiences (64.7% vs. 77.0%, childhood accident, p = 0.007). However, in some respects, they are also seen more favorable than people with other types of trauma: With regard to blame, people reporting childhood sexual abuse are regarded somewhat less guilty than people reporting all other types of trauma (7.1% vs. 8.1–11.1%, all $p < 0.05$), and they are seen more competent as parents ($p = 0.035$) and in difficult jobs ($p = 0.005$), compared to someone reporting adult physical abuse.

**Table 2. Prevalence of negative and positive stereotypes about people with different traumatic experiences.**

| People who have experienced. . . (N) | | Childhood sexual abuse | Childhood physical abuse | Childhood accident | Adult physical abuse | H (df = 3) | p | Post hoc |
|---|---|---|---|---|---|---|---|---|
| . . .are harmed for the rest of their lives. (1302) | agree | 61.5 | 50.9 | 30.9 | 42.9 | 61.8 | < .001 | 1>2, 1>3, 1>4, 2>3, 3<4 |
| | undecided | 23.7 | 26.2 | 38.0 | 34.0 | | | |
| | disagree | 14.8 | 22.8 | 31.2 | 31.1 | | | |
| | Mean rank | 755.2 | 673.9 | 544.3 | 632.5 | | | |
| . . .have a higher risk of becoming a criminal. (1301) | agree | 18.2 | 21.5 | 17.7 | 16.0 | 15.8 | .001 | 2>3, 2>4 |
| | undecided | 21.9 | 26.8 | 15.0 | 19.4 | | | |
| | disagree | 59.9 | 51.7 | 67.3 | 64.6 | | | |
| | Mean rank | 655.9 | 707.8 | 615.4 | 625.1 | | | |
| . . .are to some extent guilty of what has happened to them. (1291) | agree | 7.1 | 11.1 | 9.7 | 8.1 | 19.6 | < .001 | 1<2, 1<3, 1<4 |
| | undecided | 3.4 | 7.1 | 11.2 | 16.5 | | | |
| | disagree | 89.5 | 81.7 | 79.1 | 75.5 | | | |
| | Mean rank | 596.4 | 647.1 | 660.7 | 680.3 | | | |
| . . .are unable to have a stable relationship. (1289) | agree | 24.1 | 20.6 | 19.9 | 22.2 | 10.1 | .018 | 1>3 |
| | undecided | 30.2 | 34.1 | 21.5 | 27.5 | | | |
| | disagree | 45.7 | 45.3 | 58.6 | 50.3 | | | |
| | Mean rank | 674.2 | 664.9 | 595.8 | 644.9 | | | |
| . . .are unpredictable. (1287) | agree | 17.7 | 19.7 | 15.4 | 14.3 | 7.9 | .049 | / |
| | undecided | 18.9 | 27.2 | 22.8 | 24.0 | | | |
| | disagree | 63.4 | 53.1 | 61.7 | 61.7 | | | |
| | Mean rank | 628.3 | 688.2 | 631.2 | 628.6 | | | |
| . . . must have already been vulnerable before the event, should they develop a mental illness. (1278) | agree | 17.6 | 22.5 | 18.5 | 18.4 | 5.1 | .16 | / |
| | undecided | 21.0 | 23.8 | 21.9 | 26.9 | | | |
| | disagree | 61.4 | 53.8 | 59.6 | 54.7 | | | |
| | Mean rank | 613.8 | 667.0 | 626.1 | 651.0 | | | |
| . . .are able to have good friendships. (1305) | agree | 69.8 | 61.4 | 78.6 | 68.9 | 23.8 | < .001 | 2<3, 3>4 |
| | undecided | 23.5 | 28.6 | 17.1 | 22.5 | | | |
| | disagree | 6.8 | 10.0 | 4.3 | 8.6 | | | |
| | Mean rank | 654.9 | 598.0 | 713.2 | 646.2 | | | |
| . . .are just as suitable for a responsible job like any other person. (1310) | agree | 83.3 | 76.6 | 72.8 | 71.7 | 13.8 | .003 | 1>3, 1>4 |
| | undecided | 11.2 | 16.6 | 20.2 | 20.7 | | | |
| | disagree | 5.5 | 6.8 | 7.0 | 7.6 | | | |
| | Mean rank | 701.3 | 658.5 | 634.7 | 627.4 | | | |
| . . .perform their parental duties just as well as other people. (1297) | agree | 76.1 | 70.8 | 74.7 | 66.3 | 9.3 | .026 | 1>4 |
| | undecided | 17.1 | 23.6 | 19.4 | 24.6 | | | |
| | disagree | 6.8 | 5.6 | 5.9 | 9.1 | | | |
| | Mean rank | 674.1 | 644.5 | 667.5 | 610.6 | | | |

(*Continued*)

**Table 2.** (*Continued*)

| People who have experienced. . . (*N*) | | Childhood sexual abuse | Childhood physical abuse | Childhood accident | Adult physical abuse | *H* (*df* = 3) | *p* | Post hoc |
|---|---|---|---|---|---|---|---|---|
| . . .have survived a crisis and have grown through it. (1301) | agree | 64.7 | 69.3 | 77.0 | 68.1 | 11.5 | .009 | 1<3 |
| | undecided | 29.4 | 24.5 | 18.1 | 26.1 | | | |
| | disagree | 5.9 | 6.1 | 4.9 | 5.8 | | | |
| | Mean rank | 619.4 | 647.3 | 696.7 | 640.3 | | | |

Results of the Kruskal-Wallis rank sum test (*H*-statistic and *p* value) and significant pairwise comparisons (Conover's post hoc test with correction for multiple testing per item using Holm's procedure).

People who suffered childhood physical abuse are seen at greater risk of committing criminal offenses (endorsed by 21.5%), a significant difference to childhood accident and adult physical trauma (16.0–17.7%, $p < 0.01$ for both comparisons). They are also seen less capable of having good friendships (61.4% vs. 78.6% childhood accident, $p < 0.001$). People who have had a severe accident in childhood are generally seen the least negatively, for example, only 30.9% (compared to 42.9–61.5%, all $p < 0.01$) see them as permanently harmed, and 78.6% (compared to 61.4–69.8%) trust them to be able to have good friendships.

## Evaluating stigma measures between scenarios (H1-3): Social distance

The desire for social distance did not show large differences between the four scenarios, and followed an opposite pattern: it was lowest towards a person having experienced childhood sexual abuse (median 1.83, interquartile range IQR = 1.17) and highest towards a person having suffered from adult physical abuse (median 2.17, IQR = 1.17), with childhood accidental trauma and childhood physical abuse (median 2.00, IQR = 1.17 each) positioned in the middle. Kruskal-Wallis rank sum test confirmed an overall difference between vignettes (H = 17.1, p < 0.001). Conover's test for multiple comparisons of mean rank sums (with Holm's correction) confirmed that social distance was significantly lower in childhood sexual abuse compared to childhood physical abuse (p = 0.019), childhood accidental trauma (p = 0.019) and adult physical trauma (p < 0.001).

## Evaluating stigma measures between scenarios (H1-3): Reluctance to address CT in conversation

Asked whether they would talk to their neighbor about their traumatic experience, 45.4% indicated this was unlikely in the case of child sexual abuse, 38.2% in the case of child physical abuse, 30.6% in the case of a childhood car accident, and 24.5% in the case of adult physical abuse (Fig 1).

Regarding the reluctance to address CT in conversation with trauma victims, Kruskal-Wallis rank sum test found significant differences between vignettes (H(3) = 40.5, p < 0.001). Conover's post hoc test (with Holm's correction) found partial confirmation of our hypotheses: While the stigma surrounding childhood sexual abuse (mean rank = 591.8) and childhood physical abuse (mean rank = 605.0) did not differ significantly (H1: p = .63), both childhood sexual and physical abuse were surrounded by greater stigma than childhood accidental trauma (mean rank = 684.3; H2: $p$ = 0.003 and $p$ = 0.011), and both childhood sexual and physical trauma were a more frequently avoided topic than adult physical trauma (mean rank = 745.7, H3: $p < 0.001$ for each comparison). Fig 1 suggests that the main difference

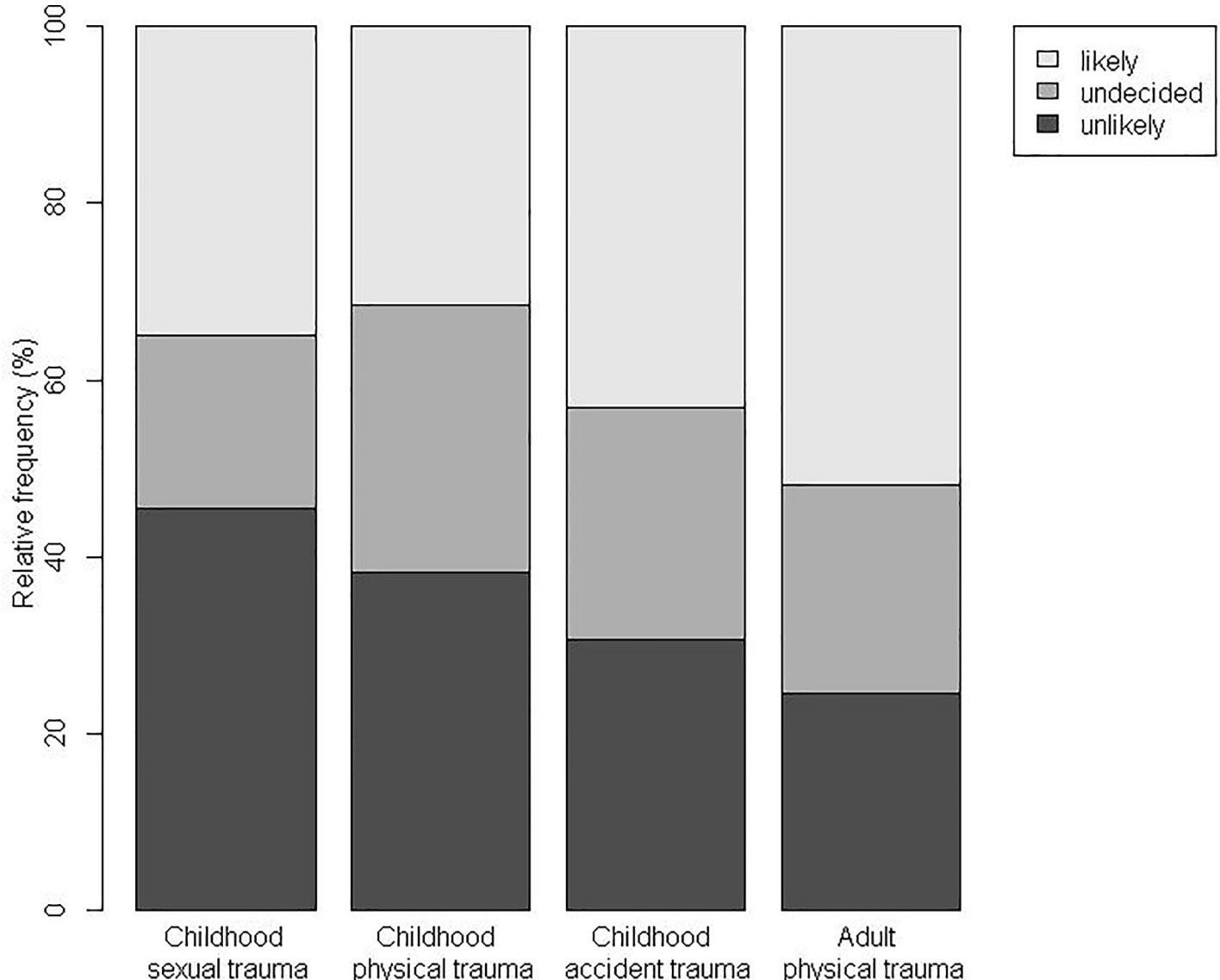

**Fig 1. Readiness to address trauma when talking to a person with traumatic experiences.** Percentage of respondents willing, undecided or unwilling to address trauma when talking to a neighbor with traumatic experiences. Representative population survey, $N = 1313$.

between childhood sexual and physical abuse is that sexual abuse elicits fewer neutral responses, corresponding to slightly more people being both ready and reluctant to talk about childhood sexual compared to physical trauma.

## Mediation analysis: Disentangling stereotypes, social distance, and reluctance to address CT in conversation (H4)

We used a single-group SEM (SGSEM) to test the interrelation of the stigma components in the entire sample, and exploratively used multi-group SEM (MGSEM) to test these interrelations for each vignette.

**Global model fit.** For both the SGSEM and MGSEM, the inferential $\chi^2$ statistic detected significant differences between the theoretical and empirical covariance matrices (all $p <$

.001), rejecting the exact-fit hypothesis (see S2 Table). However, the $\chi^2/df$-Ratio, which accounts for model complexity, if not for sample size (c.f. [32]), descriptively suggested an almost acceptable model fit for the SGSEM ($\chi/df$ = 3.01) and a good model fit for the MGSEM ($\chi/df$ = 1.65).

The RMSEA index confirmed the close-fit hypothesis for the SGSEM ($H_0$: RMSEA $\leq$ .05, $p$ = 1.00) and rejected the not-close-fit hypothesis (upper bound of the 90% confidence interval < .05), indicating that the model fitted the data well. For the MGSEM, in contrast, the RMSEA only confirmed the close-fit hypothesis ($p$ > .93) and failed to reject the not-close-fit hypothesis (upper bound of the 90% confidence interval = .051) suggesting insufficient power for this particular analysis. The SRMR was always smaller than .05, speaking in favor of a good fit for the SGSEM and MGSEM to the data. But all CFI values fell short even of the more liberal $\geq$ .95 threshold, raising doubt about its incremental fit relative to the independence (null) model.

In summary, the global fit indices only partially supported the theorized mediation model, both in the single-group and the multi-group version. We noted however, that the descriptive fit indices changed by less than .005 after listwise deletion of missing values, suggesting negligible sensitivity of the models to the missing data exclusion procedure (see S2 Table).

**Local model fit.** The residual variance-covariance matrices for the SGSEM and MGSEM can be found in the S3a-S3e Table. Since all indicators had a 5-point metric, the unstandardized values were directly comparable. For the SGSEM relatively large positive covariance residuals (< .50) remained among the social distance items, indicating that the model underestimated their interrelationships. It additionally underestimated the association between the negative stereotype items "unpredictable" and "criminal risk". The highest modification index (7.0 $\leq$ MI $\leq$ 13.3), updated after each modification, repeatedly flagged the social distance item "accept as work colleague" and the negative stereotype item "predisposed". Deleting these items from the model resulted in an acceptable CFI value of .96 for the SGSEM model. The perfect fit hypothesis remained to be rejected, though ($\chi^2$ = 164.1, $p$ < .05).

In the MGSEM analysis relatively large positive residual covariances (>.20) resulted for the negative stereotype items "predisposed" and "relationships" in the childhood sexual abuse group, as well as the social distance items "subtenant" and "child care" both in the childhood accidental trauma group and the adult physical abuse group. These relationships were, however, not among the most influential as indicated by the (iteratively updated) modification indices. In most cases these, again, were related to the social distance item "accept as work colleague" and they were located in the three childhood trauma vignette groups (7.9 $\leq$ MI $\leq$ 11.0). Neither deletion of this item nor allowing its residuals to be correlated with other item residuals in the affected groups raised the CFI above the threshold of acceptable fit (all CFI < .95) or reduced the model Chi substantially (all $p$ < .05).

In summary, the local model fit analysis identified two potentially redundant items–the social distance item "accept as work colleague" and the negative stereotype item "predisposed". After their exclusion, the descriptive global fit indices unanimously supported an acceptable to good model fit for the SGSEM. For the MGSEM no indicator constellation was found that produced consistent descriptive fit indices.

**Mediation analysis.** According to the SGSEM, negative stereotypes about trauma victims were associated with an increased general desire for social distance from such persons ($B$ = .17, $p$ < .001), which, in turn, was related to a decreased inclination to talk to an afflicted individual ($B$ = -.30, $p$ < .001). The resulting mediation effect, linking negative stereotypes to greater reluctance to talk to trauma victims (hypothesis 4), was statistically significant ($B$ = -.05, $p$ < .001; see Table 3). However, there was also a much stronger and unexpected direct effect of agreement with negative stereotypes, associated with lower reluctance to talk to trauma victims ($B$ = .15, $p$ < .001). These patterns were replicated after exclusion of the two potentially

**Table 3. Results of the mediation analysis.**

| Model | Model specifics | Vignette | Negative stereotypes | | | Positive stereotypes | | | R² (Reluc-tance) |
|-------|----------------|----------|------------|----------------|------------|-----------|----------------|-----------|------|
| | | | Direct (d) | Mediation (a*c) | Total (d +a*c) | Direct (e) | Mediation (b*c) | Total (e +b*c) | |
| SGSEM | Pairwise deletion of missings | | .15*** | -.05*** | .09* | -.05 | .18*** | .13** | .06 |
| | Listwise deletion of missings | | .16*** | -.05** | .11* | -.02 | .15*** | .13** | .05 |
| | Deletion of "predis-posed" & "colleague" | | .13** | -.03* | .10* | -.02 | .16*** | .14** | .06 |
| MGSEM | Pairwise deletion of missings | CST | .27** | -.14* | .13 | -.15 | .26** | .12 | .14 |
| | | CPT | .25** | -.05 | .20* | .00 | .14** | .14 | .08 |
| | | CAT | .21* | -.00 | .21* | .23* | .02 | .25** | .06 |
| | | APT | -.02 | -.07 | -.09 | -.30* | .42*** | .12 | .19 |
| | Listwise deletion of missings | CST | .25* | -.13* | .12 | -.20 | .21* | .01 | .11 |
| | | CPT | .29** | -.04 | .25* | .16 | .09 | .25** | .10 |
| | | CAT | .16 | -.01 | .15 | .18 | .03 | .22** | .05 |
| | | APT | .05 | -.06 | -.01 | -.31 | .48** | .17 | .15 |

Standardized path coefficients (Beta values) and significance level (*p*) of the mediation analysis in the full sample (single-group structural equation modelling—SGSEM) and with vignette as group factor (multi-group—MGSEM) using a weighted least squares estimator and listwise or pairwise deletion of missing values, respectively. For the path coefficient labels a–d please refer to S1 Fig.

*$p < .05$

** $p < .01$

*** $p < .001$.

redundant items identified in the local model fit analysis. In sum, and in contrast to hypothesis 4, the beneficial direct effect prevailed ($B = .09$, $p = .022$), suggesting that people endorsing negative stereotypes about trauma victims in general, still tend to ask an afflicted individual about his or her experiences, if they are invited just like in our neighbor scenario.

The MGSEM analysis revealed that the effects of negative stereotypes were driven primarily by the childhood interpersonal trauma groups. Specifically, in both groups we replicated the unexpected positive correlation between negative stereotypes and the willingness to talk to trauma victims ($B = .27$, $p = .006$ and $B = .25$, $p = .008$; see Table 3 and Figs 2 and 3). Both groups also supported the predicted pathway from negative stereotypes to the desire for social distance (childhood sexual abuse: $B = .28$, $p = .006$ and childhood physical abuse: $B = .21$, $p = .017$, respectively). However, only the childhood sexual abuse group also showed a robust relationship from the desire for social distance to the reluctance to talk ($B = -.49$, $p < .001$). The resulting mediation effect of negative stereotypes reached significance in this group ($B = -.14$, $p = .018$), and here it blocked the positive effects negative stereotypes have on the willingness to talk to a trauma victim (total effect: $B = .13$, $p = .14$). In the light of the inconsistent global fit indices for the MGSEM, this block should be considered as 'partial', as the non-significance of the total effect may be related to an insufficient power of this particular test. In contrast, in the childhood physical abuse vignette group the unexpected direct effect predominated (total effect: $B = .20$, $p = .026$) and the childhood accidental trauma and adult physical abuse vignette groups yielded no robust pathways between negative stereotypes and the willingness to approach trauma victims. Our measurement invariance analysis showed an unusual pattern, with an initial *increase* of ΔCFI from configural to metric invariance and decrease from there to scalar invariance (see S4 Table). However, the total loss in CFI from configural to scalar invariance (-.009) was smaller than the generally accepted threshold of ΔCFI = -.01, and both ΔRMSEA and ΔSRMR increased by less than their respective thresholds (ΔRMSEA = 0.00,

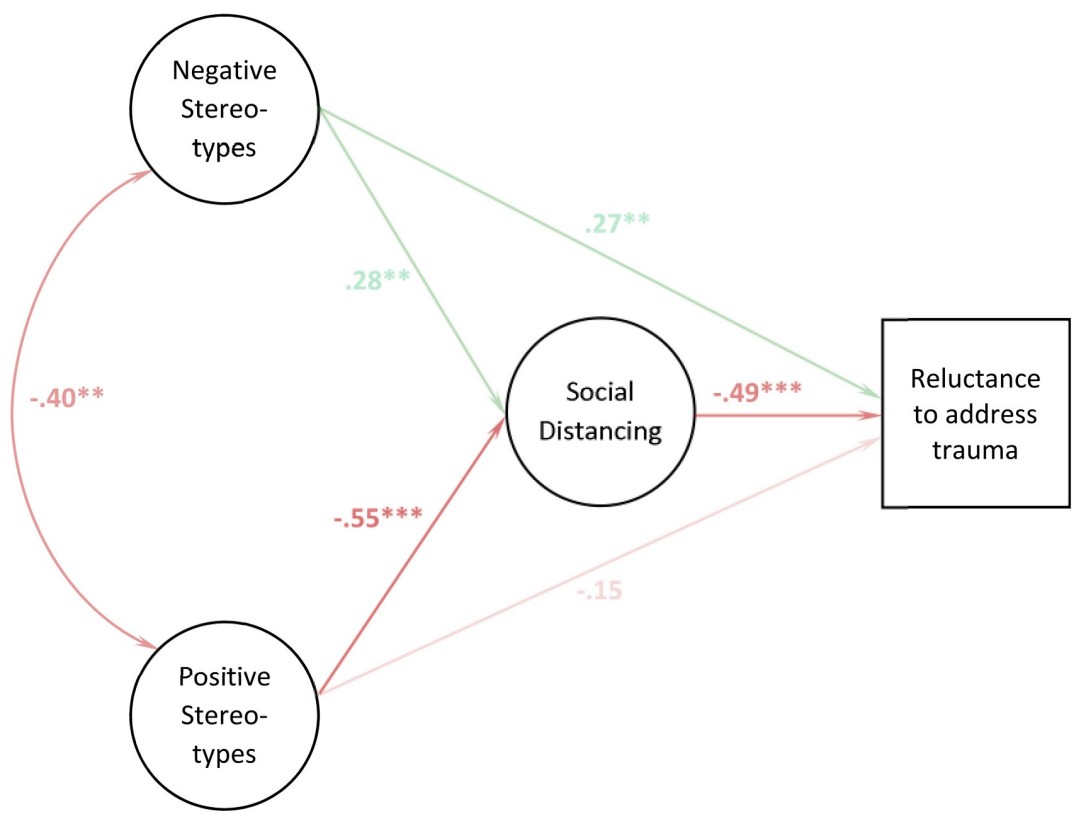

**Fig 2. Path coefficients of the MGSEM analysis for vignette "childhood sexual trauma".** Weighted least squares estimated path coefficients (standardized). The parameter estimates are represented by the opacity level of the arrows.

$\Delta$SRMR = .01). The $\Delta\chi^2$ value for the scalar invariance test was significant but over-powered due to the large sample size. We therefore believe that the (descriptively) different association patterns in the four groups indeed may represent different stigmatization processes, rather than psychometric instability of the three constructs.

The SGSEM results further revealed that positive stereotypes are accompanied by an increased tendency to talk to trauma victims (total effect: $B$ = .13, $p$ = .001) and that this relationship was driven largely by a reduced desire for social distance from such persons (mediation term: $B$ = .18, $p < .001$). However, no clear pattern emerged for the corresponding MGSEM analysis. Positive stereotypes indeed were associated with a reduced social distance in all four study groups ($B \leq$ -.53, all $p < .001$) but the mediation term was robustly significant only in the childhood sexual abuse and adult physical abuse vignette groups ($B$ = .26, $p$ = .004 and $B$ = .42, $p < .001$, respectively), where it was too weak to translate to real life behavior (total effects: both p > .159). In childhood accidental trauma group, on the other hand, we observed a robustly increased overall willingness to talk to a certain victim under the influence of positive stereotypes (total effect $B$ = .25, $p$ = .001) but it lacked a robust direct or indirect pathway as foundation.

## Discussion

As the first comprehensive study on public stigma of CT in a population sample, this paper examined three aspects of stigma towards persons with different types of CT, and their interrelations. Unexpectedly, childhood sexual abuse did not consistently elicit more stigma than

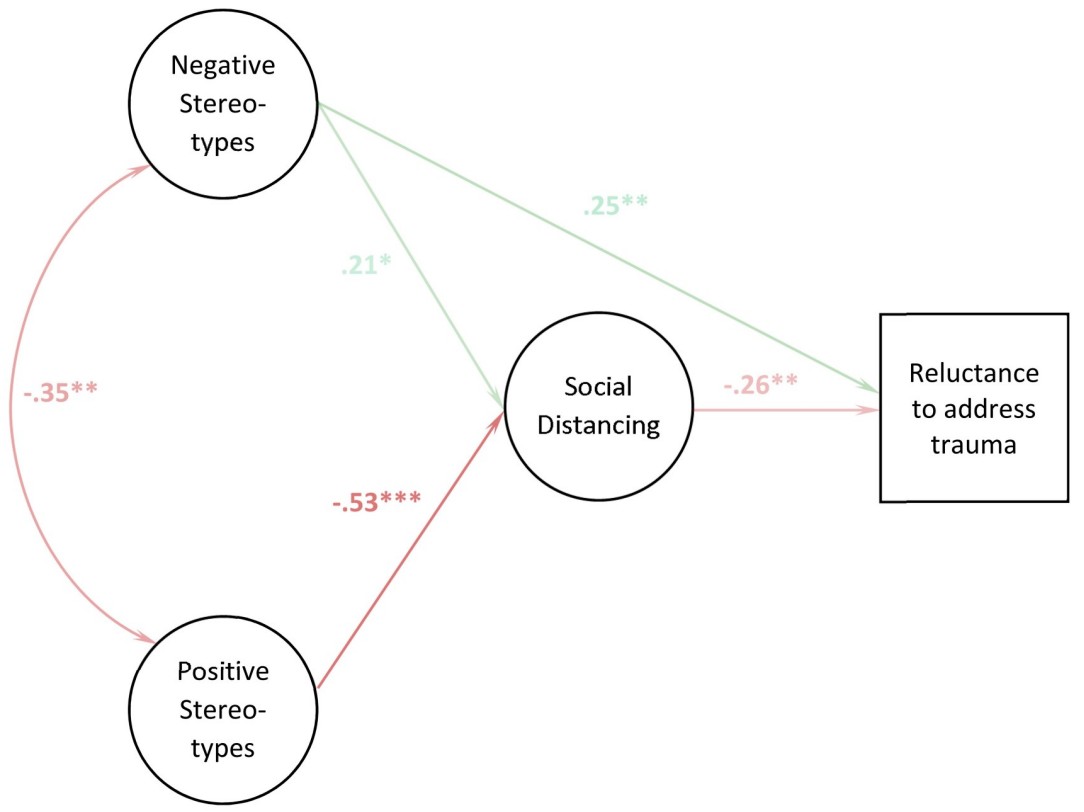

**Fig 3. Path coefficients of the MGSEM analysis for vignette "childhood physical trauma".** Weighted least squares estimated path coefficients (standardized). The parameter estimates are represented by the opacity level of the arrows.

childhood physical abuse (H1). In line with our hypothesis H2, negative attitudes and reluctance to address CT in conversation were more pronounced toward persons with interpersonal childhood trauma (childhood sexual and physical abuse) than toward persons with childhood accidental trauma, while social distance was lower toward persons with childhood sexual abuse than toward persons with childhood accidental trauma, which was contrary to our expectations. Comparing childhood physical with adult physical trauma (H3), here our results were equally mixed, suggesting that CT does not generally elicit more stigma than adult trauma, at least with regard to physical trauma as tested in this study. Finally, when examining the interrelations between stereotypes, social distance, and reluctance to address CT in conversation (H4) we found that social distance was a mediator of the relationship between negative stereotypes and reluctance to reach out to CT trauma survivors.

In descriptive analyses we found only one in three respondents willing to reach out to someone indicating they were still struggling with interpersonal childhood trauma. People were particularly reluctant to talk about childhood sexual and physical trauma, as opposed to childhood accidental trauma and adult physical trauma. Although people in the vignette mentioned their traumatic experience, and hinted at ongoing related difficulties, only one in three respondents considered actively raising this topic with them again, compared to one in two in adult physical trauma. Our results thus confirm the difficulties surrounding conversations about CT experiences among the general public.

In the current study, the majority of respondents (61.5%) agreed with the statement that persons with childhood sexual abuse would be permanently damaged. These findings are in

line with a study by Finkelhor [22] in (non)victims of childhood sexual abuse, where more than 65% of victims and non-victims stated that childhood sexual abuse has great permanent effect. Interestingly, in contrast to previous studies who found greater social distance toward survivors of childhood sexual abuse [19, 20], in the current study childhood sexual abuse provoked the least desire for social distance. One possible explanation is that the aforementioned studies were conducted among professional groups with professional contact to survivors of childhood sexual abuse, whereas the current study was conducted in the general population. However, negative and positive stereotypes showed a plausible pattern in the current study, with childhood physical trauma being associated with lower expectations regarding friendships, and stronger suspicion of criminal offenses, and childhood sexual trauma evoking also difficulties in partnerships. Hence, there are indications that a stigma of CT exists, and that it does in fact to the reluctance to interact with survivors of CT.

The SGSEM analysis revealed that negative stereotypes indeed affect the reluctance to address CT in conversation with trauma victims and that one relevant mediator of this relationship is the desire for social distance. In line with our fourth hypothesis, the result of this mediation process is adverse. That is, by increasing the desire for social distance, negative stereotypes increase reluctance to address CT in conversation with trauma victims. Surprisingly however, this effect is mitigated by an unexpected direct pathway representing an increased willingness to talk to trauma victims even in participants that entertain negative stereotypes about such victims. Overall, our initial model showed only partial fit to the data, but it proved amenable to improvements. Erasing two potentially redundant indicators raised all descriptive fit indices above generally accepted thresholds and the predicted pathways remained stable. We have to acknowledge, however, that, in the statistical sense (model Chi), the model failed to adequately represent the data and that the explained variance of our outcome "reluctance to address trauma in conversation", was small (6%).

In line with our hypotheses 1–3 our exploratory MGSEM suggested, that the strongest negative effects of negative stereotypes were related to childhood sexual trauma. Here the adverse mediated and advantageous direct effect of negative stereotypes partially cancelled each other out. In childhood physical trauma, in contrast, the unexpected direct positive effect of negative stereotypes stood isolated, resulting in an overall increased willingness to ask trauma victims about their experiences. The surprising direct effects in both groups suggest as yet unidentified mechanisms and motives associated with negative stereotypes of childhood interpersonal trauma victims. The theorized model poorly explained our data acquired with the childhood accidental trauma-vignette and, surprisingly, also with the adult physical trauma scenario. We expect that this fact lead to the unsatisfactory global fit indices for the MGSEM.

Thus, the SEM analysis revealed the complexity of reactions towards persons with CT. The most likely explanation for these findings is that beyond stigma, other motives guide the willingness to talk about traumatic events in conversation with trauma victims. Possibly, endorsing negative stereotypes is also related to notions like taking the problem seriously and not trivializing the consequences of childhood abuse. Then, endorsing negative stereotypes could be related to worrying about the person or pitying them, which could correspond to greater openness to talk about what had happened to them.

To our knowledge, this is the first comprehensive study examining public stigma with regard to CT. Considering its strengths and limitations, the lack of a measure of positive and negative emotions is probably its most severe shortcoming. Theoretically, positive emotions like empathy could also be related to greater reluctance to address trauma, reflecting a desire to protect the person or to avoid hurting them by raising a painful topic. Examining the interplay of emotions, and the different aspects of stigma when dealing with reports of CT survivors, is thus a desideratum for future research. In addition, the vignette scenario, and the

subsequent elicitation of respondents' reluctance to address CT in conversation with a trauma victim might be culturally sensitive. That is, while it is generally common to reach out to neighbors in Germany, in other cultural contexts it might be considered impolite, limiting the transferability of the vignette scenario to other cultural contexts. Further, due to limited resources and funding, we were only able to include one type of adult trauma, so that any difference between reactions to an adult or childhood trauma victim are elicited with regard to physical, but not sexual abuse. Among the strengths of our study are its representative general population sample and the use of the short scenarios of an identical situation pertaining to four different types of trauma, which enabled us to compare public reactions to these different types.

Our study fills an obvious gap in trauma related stigma research. So far, the stigma associated with trauma has mainly been explored with regard to stigma experience rather than public expressions of stigma. Moreover, previous research on the stigma of childhood abuse has put a strong focus on survivors of childhood sexual trauma. We found survivors of childhood physical abuse stigmatized to a similar degree, evoking even higher desire for social distance compared to survivors of childhood sexual abuse. This group of victims has so far received considerably less attention in stigma research.

Given the role of social support in preventing adverse long-term outcomes of childhood abuse [41, 42], and the role of shame, stigma and devaluation in further victimizing the survivors of childhood abuse, focusing on the stigma of being traumatized itself seems justified. It should be noted, however, that in many survivors of childhood adversities, multiple stigmatizing conditions might overlap, like having developed a mental illness or substance use disorder [43], leading to intersectional stigma [44].

Improving the way we interact with victims of CT could ameliorate the adverse outcomes of trauma. However, our study shows that stigma might not be sufficient to explain hesitation to engage with trauma victims. The interplay of negative stereotypes, social distance, fear of causing new harm, emotions and desire to help seems thus a valuable subject for future studies. Understanding why people are reluctant to talk to someone who signals they are in need for help for past traumatic experiences will pave the way for interventions that increase social support for people with a history of CT.

## Supporting information

**S1 Fig. Mediation model specification.** Structural equation model specification for the proposed relationship between stereotypes, the desire for social distance, and the willingness to talk to a trauma victim. The measurement part (left) shows the two exogenous latent predictors ($\xi_1, \xi_2$, operationalized by exogenous indicators x1-x9 in the same the order as S1 Table) and the endogenous latent mediator ($\eta_1$, operationalized by exogenous indicators y1-y6). Arrows represent parameters (measurement errors $\delta_1$-$\delta_9$ or $\varepsilon_1$-$\varepsilon_6$, factor loadings—or -, and factor covariance $\phi_{12}$)–dashed if they were fixed. The structural part (right) shows the endogenous criterion (y7), its measurement error ($\varepsilon_7$), the standardized path coefficients relating it to the exogenous ($\gamma_1$-$\gamma_4$) and endogenous ($\beta_1$) latent variables (labels a-d according to Table 3 in the manuscript), and the residuum of the mediator ($\zeta_1$).
(TIF)

**S1 Table. Rotated factor loadings of stereotype items.** Results of the principal components analysis with varimax rotation and without Kaiser-normalization. $h^2$ communality.
(DOCX)

**S2 Table. Global model fit of the mediation model.** Global model fit indices of the mediation model in the full sample (single-group structural equation modeling—SGSEM) and with

vignette as group factor (multi-group—MGSEM) using a weighted least squares mean- and variance-adjusted estimator and pairwise or listwise deletion of missing values, respectively. A modified variant of the SGSEM, without the indicators "must have been vulnerable already" and "accept as colleague" is also shown. CFI comparative fit index, CI confidence interval, *df* degrees of freedom, *N* sample size, RMSEA root mean square error of approximation, SRMR standardized root mean square residual.
(DOCX)

**S3 Table.** a. Residual variance-covariance matrix for the SGSEM. Unstandardized residuals of the mediation model in the full sample using single-group structural equation modeling (SGSEM) with a weighted least squares mean- and variance-adjusted estimator and pairwise deletion of missing values. b. Residual variance-covariance matrix for the MGSEM–childhood sexual trauma subgroup. Unstandardized residuals of the childhood sexual trauma vignette group estimated using a multi-group structural equation model (MGSEM) with a weighted least squares mean- and variance-adjusted estimator and pairwise deletion of missing values. c. Residual variance-covariance matrix for the MGSEM–childhood physical trauma subgroup. Unstandardized residuals of the childhood physical trauma vignette group estimated using a multi-group structural equation model (MGSEM) with a weighted least squares mean- and variance-adjusted estimator and pairwise deletion of missing values. d. Residual variance-covariance matrix for the MGSEM–childhood accident trauma subgroup. Unstandardized residuals of the childhood accident trauma vignette group estimated using a multi-group structural equation model (MGSEM) with a weighted least squares mean- and variance-adjusted estimator and pairwise deletion of missing values. e. Residual variance-covariance matrix for the MGSEM–adult physical trauma subgroup. Unstandardized residuals of the adult physical trauma vignette group estimated using a multi-group structural equation model (MGSEM) with a weighted least squares mean- and variance-adjusted estimator and pairwise deletion of missing values.
(DOCX)

**S4 Table. Measurement invariance tests of the measurement model.** Global fit indices for the measurement model (i.e. latent variables and their indicator variables) and their change (Δ) when the factor loadings (metric invariance model) and intercepts (scalar invariance model) of the indicator variables are constrained to be equal across all four groups. For comparability with the full multi-group structural equation model a weighted least squares estimator and pairwise deletion of missing values was used. CFI comparative fit index, RMSEA root mean square error of approximation, SRMR standardized root mean square residual. *** $p <$ .001.
(DOCX)

## Author Contributions

**Conceptualization:** Georg Schomerus, Theresia Rechenberg, Tobias Gfesser, Sven Speerforck.

**Data curation:** Georg Schomerus, Theresia Rechenberg, Mario Liebergesell.

**Formal analysis:** Stephanie Schindler, Mario Liebergesell, Christian Sander.

**Investigation:** Theresia Rechenberg.

**Methodology:** Stephanie Schindler, Hans J. Grabe, Mario Liebergesell, Christian Sander.

**Project administration:** Georg Schomerus.

**Software:** Stephanie Schindler.

**Supervision:** Georg Schomerus, Hans J. Grabe, Sven Speerforck.

**Validation:** Christine Ulke.

**Writing – original draft:** Georg Schomerus, Stephanie Schindler, Theresia Rechenberg, Christine Ulke.

**Writing – review & editing:** Georg Schomerus, Stephanie Schindler, Theresia Rechenberg, Tobias Gfesser, Hans J. Grabe, Mario Liebergesell, Christian Sander, Christine Ulke, Sven Speerforck.

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
