## [Decision Letter · Decision Letter 0]

27 Apr 2021

PONE-D-21-09346

Stigma as a barrier to addressing childhood trauma in conversation with trauma survivors: a study in the general population

PLOS ONE

Dear Dr. Schomerus,

Thank you for submitting your manuscript to PLOS ONE. After careful consideration, we feel that it has merit but does not fully meet PLOS ONE’s publication criteria as it currently stands. Therefore, we invite you to submit a revised version of the manuscript that addresses the points raised during the review process.

We look forward to receiving your revised manuscript.

Kind regards,

Astrid M. Kamperman

Academic Editor

PLOS ONE

Journal Requirements:

Reviewers' comments:

Reviewer's Responses to Questions

**Comments to the Author**

1. Is the manuscript technically sound, and do the data support the conclusions?

Reviewer #1: Partly

Reviewer #2: Partly

2. Has the statistical analysis been performed appropriately and rigorously? 

Reviewer #1: No

Reviewer #2: No

3. Have the authors made all data underlying the findings in their manuscript fully available?

Reviewer #1: No

Reviewer #2: Yes

4. Is the manuscript presented in an intelligible fashion and written in standard English?

Reviewer #1: Yes

Reviewer #2: Yes

5. Review Comments to the Author

Reviewer #1: The current paper confronts an important social issue, convincingly describing the problem of stigma around child abuse in the introduction. The results tell a convincing story which are interpreted correctly, accessibly and in detail in the discussion. The authors deserve praise for their work in those respects.

However, there are aspects in methods section that beg more explanation than has been provided in my opinion. I am attaching an annotated PDF which highlights some specific concerns in the methods section. Those concerns can broadly be grouped in three points:

First, a lot of information is removed from the data via dichotomization. Likert scales are combined in categories, which in one instance is further reduced to a yes/no variable. Although it is clear that the intention is to ease the burden on the reader and provide a more straightforward analysis, the problems with dichotomizing without a specific rationale have been well known for a while (e.g. Cohen’s 1983 The Cost of Dichotomization). My intention is not to say that the decision was wrong, merely that the costs incurred from such procedures should be explicitly justified, or even empirically quantified via sensitivity analyses.

Second, and perhaps most importantly, none of the instruments used are described in terms of reliability and validity. This is especially important if novel tools are used for this study. The results could be considered by some outright meaningless if we do not know that the right thing is measured and how well it is measured.

Third, more information on missing data should be provided. The authors nicely provide a comparison between the current sample and the total German population, which is very much appreciated. However, they also mention that some respondents are not included if a certain proportion of responses is missing. Furthermore, since regressions are used extensively and no information is given about handling missing covariates, I assume that incomplete cases are dropped from these regression analyses. At minimum it is important to include how many people were dropped and how many were included in each analysis. Once step further would be to compare the dropped versus analyzed groups (perhaps in the supplementary materials), in a descriptive manner similar to the earlier comparison between the current sample and the total German population.

Without these issues addressed in some form I am reluctant to interpret the presented results. As a general note on the discussion, I am happy with the way the authors interpreted their findings. What is missing in almost all paragraphs is some reference to other work in the area. To build up our theoretical knowledge base of the issue, the current results should be embedded solidly in past research and findings, even if no exact study is similar to the current one. For each phenomena found and described I should be able to follow-up my understanding with other works which considered similar issues.

I believe the current manuscript can be an important contribution to the area, but for now I have to recommend major revisions due to the issues outlined above.

Sidenote - although the authors state that the data is fully accessible, I could not find the database information or link in the manuscript. It is possible that this information was accidentally omitted.

Reviewer #2: Thank you for the opportunity to review this manuscript, “PONE-D-21-09346 Stigma as a barrier to addressing childhood trauma in conversation with trauma survivors: a study in the general population.” This manuscript presents the results of a multivariate, combination correlational and experimental research study, conducted via phone interviews with a large representative sample of the German population, to examine stigmatizing reactions toward adult survivors of childhood physical and sexual abuse, both in general and in response to specific vignettes.

Strengths of this manuscript include the focus on an understudied topic of great clinical and social significance (childhood trauma); the authors’ clear rationale for studying societal stigma, due to its “aggravating” the negative, long-term developmental consequences of childhood trauma; the large representative sample; and summary of results in accessible tables and figures.

Suggested areas for strengthening the manuscript follow:

-Placing claims of the manuscript in the context of previous literature: The authors note in the abstract, introduction, and discussion that “there are no empirical studies examining public attitudes towards survivors of CT” (e.g., lines 100-101). In the U.S., at least, there are a number of empirical studies within the past 5 years that address public attitudes and stigma toward adult survivors of childhood trauma, especially childhood sexual abuse. The manuscript would be strengthened by positioning the contributions of the current study within the context of recent literature.

-Related to the above, it could be argued that the authors are studying the phenomenon of stigma toward *adult survivors of* childhood trauma, rather than childhood trauma per se. In some areas this could be clarified.

-Organization of introduction: The first paragraph was clear and then the flow of the introduction, from paragraph to paragraph, was harder to follow for the remainder of the intro.

-Clarification of study design in the abstract: In the abstract, the study design and research questions were unclear (e.g., how many conditions, between-subjects, within subjects, experimental or correlational, comparing childhood vs. adulthood trauma or otherwise etc.)

-Rationale for study design: Greater clarity is needed in the introduction for the study’s experimental design (what it is) and also for the rationale for why the different categories of trauma were chosen. The authors begin presenting their hypotheses on line 112 of the introduction, but at that point the overall study design is not yet clear. For example, the authors state “we hypothesize that interpersonal trauma is associated w/ more stigma than accidental trauma” (line 117), but at this point, it is not clear (a) what types of interpersonal trauma are being referred to here— e.g., childhood physical abuse, childhood sexual abuse, adult physical abuse?) and (b) what type of accidental trauma is being referred to, and whether it’s in childhood or adulthood.

-Related to the above questions, how did the authors choose adult physical abuse as the type of adult trauma being investigated? Why not adult sexual and physical abuse? And why not include an adult accident condition?

-More information is needed regarding the rationale for and theory- or data-driven process of designing the neighbor vignette. Has the neighbor vignette scenario been used in previous stigma research? If not, how did the authors choose a neighborly discussion as the setting for a trauma disclosure? For instance, in a U.S. context, it would be unusual for a person to immediately share with their neighbor, upon first meeting, that they are being abused or that they have a history of abuse. Thus, a negative reaction to this disclosure may not be due to stigma toward trauma survivors per se, but due to the person violating social norms around what information is appropriate to share upon first meeting. If this is the case, this would be an experimental confound that undermines the validity of the results.

-Regarding responses to the vignettes, what was the rationale for the decision to collapse the continuous response scale into only 3 categories? (lines 154-158)

-Regarding the statistical analyses and the fit between the hypotheses and analyses: Many inferential statistical tests are performed in this study, inflating the potential for a Type I error and diluting the conceptual clarity of the study. Manuscript clarity and contribution to the literature can be enhanced by a more streamlined connection between research questions, hypotheses, and data analyses.

-Interpretation of results in the Discussion would benefit from being tied more closely to the results. E.g., the authors state on the first line of the Discussion, “we found considerable reluctance to reach out to someone indicating they were still struggling with traumatic childhood experiences.” “Considerable reluctance” is not an accurate statement in that the majority of the sample was willing to reach out.

Smaller areas in need of clarification or adjusting:

-“Social distance / distancing” - in the U.S. this term has been used over the past year to refer to the need to socially distance from others in order to prevent infection with the coronavirus. Use of this term may be confusing or distracting for U.S. readers.

-Consistency of terminology - why “stigma” in some areas and “taboo” in others?

-“Abuse specific stigmatization” (line 85)- please clarify?

-Adjusting causal verbs used to describe pathways in the structural equation model (e.g., “increases” “decreases”) to correlational verbs (p. 16).

-For Figure 1, providing a legend that labels each scenario and labeling the Y axis.

-For Figure 2, providing numeric coefficients / parameter estimates for each pathway in the model and providing a caption or legend that labels the indices of each factor with names instead of q1, q4, q5 etc. I was not able to open the supplementary documents and anticipate that other readers would want ready access to the numeric results of the structural equation model to be able to interpret the results and understand the table fully.

6. PLOS authors have the option to publish the peer review history of their article (what does this mean?). If published, this will include your full peer review and any attached files.

Reviewer #1: **Yes: **Milan Zarchev

Reviewer #2: No

---

## [Author Response · Author response to Decision Letter 0]

11 Jun 2021

Answers to the reviewers

We thank both reviewers for their insightful comments. We have rewritten large parts of the manuscript and adapted our analytical strategy, now reporting descriptive analyses and structural equation modelling to test our hypotheses 1-4. This has greatly helped to clarify our intentions and the results of this study. We really appreciate the careful and friendly look at our paper and hope that the reviewers agree with the changes we made. Please find our answers to each of the points raised below.

Reviewer #1: The current paper confronts an important social issue, convincingly describing the problem of stigma around child abuse in the introduction. The results tell a convincing story which are interpreted correctly, accessibly and in detail in the discussion. The authors deserve praise for their work in those respects. However, there are aspects in methods section that beg more explanation than has been provided in my opinion. I am attaching an annotated PDF which highlights some specific concerns in the methods section. Those concerns can broadly be grouped in three points:

First, a lot of information is removed from the data via dichotomization. Likert scales are combined in categories, which in one instance is further reduced to a yes/no variable. Although it is clear that the intention is to ease the burden on the reader and provide a more straightforward analysis, the problems with dichotomizing without a specific rationale have been well known for a while (e.g. Cohen’s 1983 The Cost of Dichotomization). My intention is not to say that the decision was wrong, merely that the costs incurred from such procedures should be explicitly justified, or even empirically quantified via sensitivity analyses.

Response: Your point is well taken. We now no longer use binary outcomes (we omitted all logistic regressions) and use the original 5-point variables as predictor and outcome variables in structural equation modeling. See also first comment to the annotated pdf below.

Second, and perhaps most importantly, none of the instruments used are described in terms of reliability and validity. This is especially important if novel tools are used for this study. The results could be considered by some outright meaningless if we do not know that the right thing is measured and how well it is measured. 

Response: Thank you for this valuable point. We now describe the instruments in terms of validity and internal consistency reliability. We further described the stigma measures in greater detail in the introduction (p.4-5).

Social Distance Scale (primary outcome): Information was added regarding internal consistency reliability and construct validity (p.7). 

Vignette component addressing CT in conversation: Information was added about the use of vignettes in survey research and their enhancement of internal validity by allowing for random allocation of participants to the individual experimental conditions (p.7). 

Childhood Trauma Screener: Information about validity and reliability was added (p.9).

Third, more information on missing data should be provided. The authors nicely provide a comparison between the current sample and the total German population, which is very much appreciated. However, they also mention that some respondents are not included if a certain proportion of responses is missing. Furthermore, since regressions are used extensively and no information is given about handling missing covariates, I assume that incomplete cases are dropped from these regression analyses. At minimum it is important to include how many people were dropped and how many were included in each analysis. Once step further would be to compare the dropped versus analyzed groups (perhaps in the supplementary materials), in a descriptive manner similar to the earlier comparison between the current sample and the total German population.

Response: We fully agree with your concerns and thank you for your suggestions. We added the number of valid responses for all major outcomes (mean social distance and reluctance to address CT on page 8, stereotypes in table 2). As no more than 3.2% responses were missing for each outcome and we already deleted all regression analyses (logistic and multiple), we refrained from a detailed analysis of the missing values. We, of course, also report the number of included subjects for the structural equation model (table S3) and employ a sensitivity analysis of its missing values procedure to reduce a potential bias by missing values.

As a general note on the discussion, I am happy with the way the authors interpreted their findings. 

Response: Thank you very much for this feedback!

What is missing in almost all paragraphs is some reference to other work in the area. To build up our theoretical knowledge base of the issue, the current results should be embedded solidly in past research and findings, even if no exact study is similar to the current one. For each phenomena found and described I should be able to follow-up my understanding with other works which considered similar issues.

Response: We appreciated this comment and took the opportunity to add references throughout the manuscript (introduction, methods, discussion). 

Sidenote - although the authors state that the data is fully accessible, I could not find the database information or link in the manuscript. It is possible that this information was accidentally omitted.

Response: We sincerely apologize for this mishap. We hope you can now readily access the raw dataset (R-file). 

Annotated pdf (reviewer 1)

-Interview and vignettes:

p.6. : we inverted the scale: ...What's the reasoning behind collapsing some of the information?

Response: Your comment was very much appreciated. As noted above, we no longer use binary outcome variables (the logistic regressions were erased). Hypotheses H1-H3, however, were analyzed using “reluctance to address CT” and the individual stereotype items after collapsing each into three categories. We note this and our motivation - readability- in the statistics section on page 10: “To reduce the information load of our report of these analyses, we collapsed the five categories of the stereotype items and the willingness to talk to a victim item into three categories representing ‘agreement’, ‘undecidedness’, and ‘disagreement’.” In order to use the available information/variances fully, the SEM now analyzes all indicators with the original (or inverted) 5-point scaling. 

-Social Distance Scale

p.7: How many had more than 2 items missing? Reporting on missing data is important 

Response: We thank you for your question. We lost 15 people for this particular outcome measure. We now report the number of valid responses for all major outcomes.

p.7 Was the scale adapted at all or just used verbatim? In any case, some mention of the validity and reliability of the scale is important, seeing as how mean scores are used.

Response: An established German version of the scale was used verbatim, referring to the person described in the vignette. As indicated above, information was added regarding internal consistency reliability and construct validity (p.7). In addition, the SEM analyses were complemented by a measurement invariance testing showing that the factor loadings of this scale and the stereotypes are reasonably consistent (reliable) across the subgroups/vignettes.

-Stereotypes: 

p.7: Are these discussions described in another paper you could reference? Or was this measurement instrument developed for this study, in which case some more information in an appendix would be desirable (e.g. how many people with lived experiences and what type of lived experiences, how structured were the discussions, etc.).

Response: The scale was developed in context of a qualitative study: “Stigma as a Barrier to Treatment for Former Residents of GDR Children’s Homes – A Qualitative Study with Members of the “Betroffeneninitiative Missbrauch in DDR-Kinderheimen” by Gfesser et al. (2020), published in Psychiatrische Praxis. The list of stereotypes was assembled in collaboration with the focus group of trauma survivors (8 childhood physical abuse survivors) of the study mentioned above, as well as three childhood sexual abuse survivors, who were psychiatric patients at Greifswald University Hospital at the time. Participating CT survivors and participating psychotherapists were asked about relevant positive and negative stereotypes in a semi-structured telephone interview. We added this information on p.9

-Stereotypes: “inverted the scale and regrouped...”: This type of decision also requires some rationale.

Response: We added the rationale on p.9: We inverted the scale so that the results may be interpreted intuitively with higher values representing stronger positive or negative stereotyping.

Reviewer #2: 

Suggested areas for strengthening the manuscript follow:

-Placing claims of the manuscript in the context of previous literature: The authors note in the abstract, introduction, and discussion that “there are no empirical studies examining public attitudes towards survivors of CT” (e.g., lines 100-101). In the U.S., at least, there are a number of empirical studies within the past 5 years that address public attitudes and stigma toward adult survivors of childhood trauma, especially childhood sexual abuse. The manuscript would be strengthened by positioning the contributions of the current study within the context of recent literature.

Response: Thank you for this valuable suggestion. We did indeed miss some important work that has been done previously. We extended our search and added references and corresponding findings throughout the introduction, methods and discussion (marked in yellow). The above-mentioned statement was rephrased (abstract: p.2 ; introduction: p.4)

Abstract: So far there is no comprehensive study examining public stigma toward adult survivors of various forms of childhood trauma, and it is unclear whether stigma interferes with reaching out to affected individuals

Introduction: However, to our knowledge, there are no comprehensive population-based studies examining public stigma towards adult survivors of CT, and the available studies do not differentiate between different types of trauma.

-Related to the above, it could be argued that the authors are studying the phenomenon of stigma toward *adult survivors of* childhood trauma, rather than childhood trauma per se. In some areas this could be clarified.

Response: Good point. We clarified this in the abstract and the introduction (marked in yellow) .

-Organization of introduction: The first paragraph was clear and then the flow of the introduction, from paragraph to paragraph, was harder to follow for the remainder of the intro.

Response: Thanks for pointing this out. We now rewrote lower sections of the introduction, to improve flow and clarity of argumentation.

-Clarification of study design in the abstract: In the abstract, the study design and research questions were unclear (e.g., how many conditions, between-subjects, within subjects, experimental or correlational, comparing childhood vs. adulthood trauma or otherwise etc.)

Response: We added the following clarification to the abstract: In a vignette study based on a representative sample of the German general population (N=1320; 47.7% male) we randomly allocated participants to brief case vignettes pertaining to past childhood sexual/ physical abuse or accidents, and adult physical abuse. Stigma was elicited by applying the Social Distance Scale, assessing respondents’ attitudes/stereotypes toward the persons in the vignette and their reluctance to address the specific trauma in conversation. While our first aim is to establish the prevalence of stigma towards persons with CT, we hypothesize that attitudes differ according to the type of trauma. Applying Krukall-Wallis tests...

-Rationale for study design: Greater clarity is needed in the introduction for the study’s experimental design (what it is) and also for the rationale for why the different categories of trauma were chosen. The authors begin presenting their hypotheses on line 112 of the introduction, but at that point the overall study design is not yet clear. For example, the authors state “we hypothesize that interpersonal trauma is associated w/ more stigma than accidental trauma” (line 117), but at this point, it is not clear (a) what types of interpersonal trauma are being referred to here— e.g., childhood physical abuse, childhood sexual abuse, adult physical abuse?) and (b) what type of accidental trauma is being referred to, and whether it’s in childhood or adulthood.

Response: Thank you for pointing this out. We extended the description of the study design in the introduction, p.4-5. We also added a section on the rationale of vignette studies under methods, p.7: By choosing these scenarios, we aimed to elicit differences in attitudes between sexual versus physical childhood abuse, interpersonal versus accidental childhood abuse, and childhood versus adult abuse. We chose “a new neighbor” to describe an encounter with a previously unknown person that is relatable to all respondents and offers some choice as to how close a future relationship with the new neighbor will be. Generally, the use of vignettes allows the random assignment of study participants to different experimental conditions, which enhances the internal validity.

-Related to the above questions, how did the authors choose adult physical abuse as the type of adult trauma being investigated? Why not adult sexual and physical abuse? And why not include an adult accident condition?

Response: Thanks for asking. Due to limited resources/ funds, we opted to restrict our study to four scenarios with sufficient sample size for each scenario. We have now explained the rationale for selecting four scenarios in more detail (see above). Adult physical abuse was used, instead of adult sexual abuse, because we assumed it to have less of a gender-bias. We added an appropriate sentence to our limitation section.

-More information is needed regarding the rationale for and theory- or data-driven process of designing the neighbor vignette. Has the neighbor vignette scenario been used in previous stigma research? If not, how did the authors choose a neighborly discussion as the setting for a trauma disclosure? For instance, in a U.S. context, it would be unusual for a person to immediately share with their neighbor, upon first meeting, that they are being abused or that they have a history of abuse. Thus, a negative reaction to this disclosure may not be due to stigma toward trauma survivors per se, but due to the person violating social norms around what information is appropriate to share upon first meeting. If this is the case, this would be an experimental confound that undermines the validity of the results.

Response: Thanks for giving us the opportunity to clarify. We added a section on the use of vignette experiments, p.7. The use of social situations and interactions with different people in one’s environment is a typical element in social science research and vignette studies. It is true that the context is culturally-sensitive, however the early stigma studies, which used similar vignettes, also used neighbor vignettes in an US-context (Link et al., 1987). However, we added a sentence under limitations. We also explain that We chose “a new neighbor” to describe an encounter with a previously unknown person that is relatable to all respondents and offers some choice as to how close a future relationship with the new neighbor will be. (p.7)

-Regarding responses to the vignettes, what was the rationale for the decision to collapse the continuous response scale into only 3 categories? (lines 154-158)

Response: Thank you very much for your question. To address its implicit concern-information loss, we deleted all analyses that used the binary version of “reluctance to address CT” (logistic regressions). Hypotheses H1-H3 still use the 3-point versions of “reluctance to address CT” and the individual stereotype items to improve readability and clarity of our report. We now note this and our motivation explicitly in the statistics section on page 10: To reduce the information load of our report of these analyses, we collapsed the five categories of the stereotype items and the willingness to talk to a victim item into three categories representing ‘agreement’, ‘undecidedness’, and ‘disagreement’. To reduce the information loss, however, the SEM (H4) now uses all indicators with the original (or inverted) 5-point metric. 

-Regarding the statistical analyses and the fit between the hypotheses and analyses: Many inferential statistical tests are performed in this study, inflating the potential for a Type I error and diluting the conceptual clarity of the study. Manuscript clarity and contribution to the literature can be enhanced by a more streamlined connection between research questions, hypotheses, and data analyses.

Response: Thanks for pointing this out. We now streamlined the manuscript, deleting correlation and multiple or logistic regression analyses and reorganized the results section; H4 was now tested using SEM analysis only. Please find the new sections under statistical analysis, p.11 and results, p.17.

-Interpretation of results in the Discussion would benefit from being tied more closely to the results. E.g., the authors state on the first line of the Discussion, “we found considerable reluctance to reach out to someone indicating they were still struggling with traumatic childhood experiences.” “Considerable reluctance” is not an accurate statement in that the majority of the sample was willing to reach out.

Response: Very good point indeed. We rewrote major sections in the discussion, tying it more closely to results, and findings from the literature. We also rephrased the particular sentence to

In descriptive analyses we found only one in three respondents willing to reach out to someone indicating they were still struggling with interpersonal childhood trauma.

Smaller areas in need of clarification or adjusting:

-“Social distance / distancing” - in the U.S. this term has been used over the past year to refer to the need to socially distance from others in order to prevent infection with the coronavirus. Use of this term may be confusing or distracting for U.S. readers.

Response: We agree, but nevertheless the desire for social distance is a long established construct in attitude research, and it would not be helpful to introduce a new name for it, since this would interfere with relating the present findings to other studies using social distance measures. Since we now explain the construct and its history in more detail, we are confident that readers can distinguish it from social distancing during the present pandemia.

-Consistency of terminology - why “stigma” in some areas and “taboo” in others?

Response: Thank you for pointing this out, we now avoid the term taboo and consistently speak of willingness or reluctance to interact with survivors of CT.

-“Abuse specific stigmatization” (line 85)- please clarify?

Response: Thanks for asking us to clarify. We now replaced the term with abuse-specific shame and self-blame, to be more precise.

-Adjusting causal verbs used to describe pathways in the structural equation model (e.g., “increases” “decreases”) to correlational verbs (p. 16).

Response: Our initial motive was to improve readability by reflecting the clear chronological structure of the theoretical model. We overlooked, however, that this undermines an entirely neutral report of the results and should be limited to the discussion. We therefore weakened our wording in the results section to communicate correlational relationships only. Thank you for pointing this out!

-For Figure 1, providing a legend that labels each scenario and labeling the Y axis.

Response: We thank you for your detailed review and apologize for the time and effort this and other oversights have caused you. Fig 1 now labels each scenario and the y-axis.

-For Figure 2, providing numeric coefficients / parameter estimates for each pathway in the model and providing a caption or legend that labels the indices of each factor with names instead of q1, q4, q5 etc. I was not able to open the supplementary documents and anticipate that other readers would want ready access to the numeric results of the structural equation model to be able to interpret the results and understand the table fully.

Response: Figure 2, being relevant but not central to the interpretation, was moved to the supplementary S2. As its caption is unwieldy already, we were reluctant to lengthen it further by 15 questionnaire items. Instead, we would like to suggest a unification of the labeling in this figure: We replaced the item labels q1-16 in the measurement part of the model by short item tags that–we expect–unambiguously identify the stereotype items formulated fully in table S1 and the social distance items listed fully in Link et al. 1987. The parameter estimates of the structural part of the model for subgroups 1 and 2 are shown in figures 2 and 3, which, we fear, you were not able to access previously. We deeply apologize for this and hope you can now download them without hindrance. To reduce information load in these figures, the factor loadings (measurement part) are not shown. Instead, measurement invariance, testing whether the factor loadings are consistent across the subgroups/vignettes, was analyzed and reported in the manuscript and in table S5.

---

## [Decision Letter · Decision Letter 1]

31 Aug 2021

PONE-D-21-09346R1

Stigma as a barrier to addressing childhood trauma in conversation with trauma survivors: a study in the general population

PLOS ONE

Dear Dr. Schomerus,

Thank you for submitting your manuscript to PLOS ONE. After careful consideration, we feel that it has merit but does not fully meet PLOS ONE’s publication criteria as it currently stands. Therefore, we invite you to submit a revised version of the manuscript that addresses the points raised during the review process.

ACADEMIC EDITOR:

The authors should reflect especially on the comments from the 2nd reviewer with regards to how the hypotheses can be addressed using the presented data.

We look forward to receiving your revised manuscript.

Kind regards,

Astrid M. Kamperman

Academic Editor

PLOS ONE

Journal Requirements:

Additional Editor Comments (if provided):

Reviewers' comments:

Reviewer's Responses to Questions

**Comments to the Author**

1. If the authors have adequately addressed your comments raised in a previous round of review and you feel that this manuscript is now acceptable for publication, you may indicate that here to bypass the “Comments to the Author” section, enter your conflict of interest statement in the “Confidential to Editor” section, and submit your "Accept" recommendation.

Reviewer #1: All comments have been addressed

Reviewer #2: Partly

2. Is the manuscript technically sound, and do the data support the conclusions?

Reviewer #1: Yes

Reviewer #2: Partly

3. Has the statistical analysis been performed appropriately and rigorously? 

Reviewer #1: Yes

Reviewer #2: I don't know

4. Have the authors made all data underlying the findings in their manuscript fully available?

Reviewer #1: Yes

Reviewer #2: Yes

5. Is the manuscript presented in an intelligible fashion and written in standard English?

Reviewer #1: Yes

Reviewer #2: Yes

6. Review Comments to the Author

Reviewer #1: The authors have admirably reworked their entire analysis to address the comments to a high standard. I am impressed by their improvements and can now suggest the acceptance of the manuscript.

Reviewer #2: I appreciate the authors' responsiveness to reviewer feedback. In particular, the hypotheses are more clearly described in the Introduction; there is more evidence of citing and engaging with existing research on childhood trauma stigma; and the rationale for the Neighbor scenario is clearer, if still confounded, to my mind, with the neighbor/trauma survivor violating social norms by disclosing on a first meeting.  Despite the authors' commendable, detailed work in revising the manuscript and explaining their methods and statistical results, there remain experimental confounds in the design of the study that cannot be addressed post-hoc.  Specifically: (1) The authors' data cannot address Hypothesis 2, because trauma type (accidental vs. interpersonal) is confounded with age at which trauma occurred (the only accidental trauma vignette is in childhood), and (2) the data cannot address Hypothesis 3 as it is currently stated, because adulthood trauma is confounded with physical abuse trauma, such that the authors are not comparing childhood vs. adult abuse per se but rather childhood sexual, physical, and accidental trauma vs. adulthood physical trauma. I appreciate the constraints on statistical power and experimental conditions that can be imposed by limited financial funding for a project, but given that, the experimental design needs to be adjusted accordingly. Additionally, it is hard to follow both the rationale for and the statistical process for testing Hypothesis 4, in that here the authors drop the issue of childhood vs. adulthood and accidental vs. impersonal traumas and examine stigma and trauma broadly as constructs in the SEM model, and they also introduce the concept of positive stereotypes which had not been mentioned previously in the introduction (until the statement of H4), making it hard to follow the thread from theory to method to data analysis.  Moreover, after adding a more detailed explanation of the statistical analyses in the Results section, as requested during the review process, it is now clearer how exquisitely complex the analyses are for H4, presenting barriers to the dissemination of the authors' work, when it is so difficult to parse the multi step process of this analysis.   I realize this assessment might be disappointing to the authors who have clearly put a lot of preparation into their manuscript. But in a society already predisposed to stigma and doubt surrounding childhood trauma, I think it's all the more important that trauma research methods be rigorously designed and consistent with principles of open science such as pre-registration of hypotheses and analysis plans. Thank you for the opportunity to review this manuscript.

7. PLOS authors have the option to publish the peer review history of their article (what does this mean?). If published, this will include your full peer review and any attached files.

Reviewer #1: **Yes: **Milan Zarchev

Reviewer #2: **No**

---

## [Author Response · Author response to Decision Letter 1]

20 Sep 2021

We thank the 2nd reviewer for his careful reading of our revision and for his thought provoking critique of our paper. We acknowledge the points the reviewer raised, but despite his/her pessimistic view of the paper, we feel confident to address these concerns.

(1) I appreciate the authors' responsiveness to reviewer feedback. In particular, the hypotheses are more clearly described in the Introduction; there is more evidence of citing and engaging with existing research on childhood trauma stigma; and the rationale for the Neighbor scenario is clearer, if still confounded, to my mind, with the neighbor/trauma survivor violating social norms by disclosing on a first meeting.  Despite the authors' commendable, detailed work in revising the manuscript and explaining their methods and statistical results, there remain experimental confounds in the design of the study that cannot be addressed post-hoc.  Specifically: 

The authors' data cannot address Hypothesis 2, because trauma type (accidental vs. interpersonal) is confounded with age at which trauma occurred (the only accidental trauma vignette is in childhood), and 

(2) the data cannot address Hypothesis 3 as it is currently stated, because adulthood trauma is confounded with physical abuse trauma, such that the authors are not comparing childhood vs. adult abuse per se but rather childhood sexual, physical, and accidental trauma vs. adulthood physical trauma. 

Response: The focus of our study is on childhood trauma. So we introduced childhood accidental trauma to contrast it with both childhood physical and sexual trauma. We have made this more clear in our introduction: 

“Further, we hypothesize that childhood interpersonal trauma is associated with more stigma than childhood accidental trauma (H2).” 

Similarly, we contrast childhood physical trauma to adult physical trauma, to see whether the age of the person in the vignette makes a difference. “Since adults are perceived as more stable in order to overcome traumatic events, while children are regarded as more vulnerable and more profoundly affected by trauma, we assume that childhood physical trauma triggers more stigma than adult physical trauma (H3).” 

While we avoid the confounding suspected by the reviewer by making our focus more specific, we acknowledge that any possible difference between perceptions of adult and child trauma survivors could be dependent on type of trauma, and have acknowledged this in our limitations section: 

“Any difference between reactions to an adult or childhood trauma victim are elicited with regard to physical, but not sexual abuse.” (p. 25)

(3) I appreciate the constraints on statistical power and experimental conditions that can be imposed by limited financial funding for a project, but given that, the experimental design needs to be adjusted accordingly. Additionally, it is hard to follow both the rationale for and the statistical process for testing Hypothesis 4, in that here the authors drop the issue of childhood vs. adulthood and accidental vs. impersonal traumas and examine stigma and trauma broadly as constructs in the SEM model,

Response: We have made now clear when stating H4 why we drop the childhood/adult distinction here for a general analysis of the stigma process with regard to trauma survivors (and use the four vignettes again in an explorative multi-group SEM afterwards, see response to (5)). 

“Finally, to see to what extent the concept of stigma applies to public reactions to trauma survivors, we assumed that in accordance with established models of stigma (15), negative stereotypes about people with trauma experience increase the general desire for social distance from such persons, which ultimately manifests as a greater reluctance to talk to them about their trauma experience (H4). “

(4) and they also introduce the concept of positive stereotypes which had not been mentioned previously in the introduction (until the statement of H4), making it hard to follow the thread from theory to method to data analysis.  

Response: We introduce the negative and positive stereotypes elicited in our methods section on p. 7. 

(5) Moreover, after adding a more detailed explanation of the statistical analyses in the Results section, as requested during the review process, it is now clearer how exquisitely complex the analyses are for H4, presenting barriers to the dissemination of the authors' work, when it is so difficult to parse the multi step process of this analysis. 

Respones: We agree that our description of our SEM is complex, and somewhat hides the pretty straight forward theoretical approach that was tested with these models. We have have outlined in our H4 that we tested the established stigma process with stereotypes, social distance and resulting discrminatory behavior (see response (3)). We now add at the beginning of the Results section reporting the SEM:

“We used a single-group SEM (SGSEM) to test the interrelation of the stigma components in the entire sample, and exploratively used multi-group SEM (MGSEM) to test these interrelations for each vignette.”

In fact, the unexpected finding that negative stereotypes were directly associated with more willingness to talk to the trauma victim justifies the explorative approach looking at each vignette, showing that this was particularly the case in childhood interpersonal trauma. By reporting these unexpected findings, our study shows that stigma is a relevant, but not exhaustive concept explaining the taboo surrounding CT. We discuss this on page 25.

(6) I realize this assessment might be disappointing to the authors who have clearly put a lot of preparation into their manuscript. But in a society already predisposed to stigma and doubt surrounding childhood trauma, I think it's all the more important that trauma research methods be rigorously designed and consistent with principles of open science such as pre-registration of hypotheses and analysis plans. Thank you for the opportunity to review this manuscript.

Response: We respectfully disagree. Our findings do not diminish or question the harm done by stigma and doubt surrounding childhood trauma, but for the first time use a representative population sample to examine and understand it. Our methodology and statistical analysis are sound, and our research gives some answers to our research questions, while also posing new, unexpected questions for future research.

---

## [Editor Report · Decision Letter 2]

6 Oct 2021

Stigma as a barrier to addressing childhood trauma in conversation with trauma survivors: a study in the general population

PONE-D-21-09346R2

Dear Dr. Schomerus,

We’re pleased to inform you that your manuscript has been judged scientifically suitable for publication and will be formally accepted for publication once it meets all outstanding technical requirements.

Kind regards,

Astrid M. Kamperman

Academic Editor

PLOS ONE
---

## [Editor Report · Acceptance letter]

8 Oct 2021

PONE-D-21-09346R2 

Stigma as a barrier to addressing childhood trauma in conversation with trauma survivors: a study in the general population 

Dear Dr. Schomerus:

I'm pleased to inform you that your manuscript has been deemed suitable for publication in PLOS ONE. Congratulations! Your manuscript is now with our production department. 

Kind regards, 

on behalf of

Dr. Astrid M. Kamperman 

Academic Editor

PLOS ONE